# Natural Infection with Highly Pathogenic Avian Influenza A/H5N1 Virus in Pet Ferrets

**DOI:** 10.3390/v16060931

**Published:** 2024-06-08

**Authors:** Anna Golke, Dawid Jańczak, Olga Szaluś-Jordanow, Tomasz Dzieciątkowski, Rafał Sapierzyński, Agata Moroz-Fik, Marcin Mickiewicz, Tadeusz Frymus

**Affiliations:** 1Department of Preclinical Sciences, Institute of Veterinary Medicine, Warsaw University of Life Sciences-SGGW, Ciszewskiego 8, 02-786 Warsaw, Poland; anna_golke@sggw.edu.pl; 2Veterinary Laboratory ANIMALLAB, Środkowa 2/4, 03-430 Warsaw, Poland; parazytologia.vet@gmail.com; 3Department of Small Animal Diseases with Clinic, Institute of Veterinary Medicine, Warsaw University of Life Sciences-SGGW, Nowoursynowska 159c, 02-776 Warsaw, Poland; tadeusz_frymus@sggw.edu.pl; 4Chair and Department of Medical Microbiology, Medical University of Warsaw, Chałubińskiego 5, 02-004 Warsaw, Poland; tdzieciatkowski@wum.edu.pl; 5Department of Pathology and Veterinary Diagnostics, Institute of Veterinary Medicine, Warsaw University of Life Sciences-SGGW, Nowoursynowska 159c, 02-776 Warsaw, Poland; rafal_sapierzynski@sggw.edu.pl; 6Division of Veterinary Epidemiology and Economics, Institute of Veterinary Medicine, Warsaw University of Life Sciences-SGGW, Nowoursynowska 159c, 02-776 Warsaw, Poland; agata_moroz@sggw.edu.pl (A.M.-F.); marcin_mickiewicz@sggw.edu.pl (M.M.)

**Keywords:** highly pathogenic avian influenza (HPAI), influenza virus type A (IAV), ferret, A/H5N1

## Abstract

The study involved five ferrets from one household in Poland, comprising three sick 9-week-old juveniles, their healthy mother, and another clinically normal adult, admitted to the veterinary clinic in June 2023. The juvenile ferrets displayed significant lethargy and a pronounced unwillingness to move with accompanying pulmonary distress. Prompted by concurrent outbreaks of A/H5N1 influenza virus infections in Polish cats, point-of-care tests were conducted that revealed type A influenza antigens in the throat swabs of all five ferrets. Despite treatment, one juvenile ferret exhibited dyspnea and neurological symptoms and eventually died. The two remaining ferrets recovered fully, including one severely affected showing persistent dyspnea and incoordination without fever that recovered after 11 days of treatment. In the RT-qPCR, the throat swabs collected from all surviving ferrets as well as the samples of lungs, trachea, heart, brain, pancreas, liver, and intestine of the succumbed ferret were found positive for A/H5N1 virus RNA. To our best knowledge, this is the first documented natural A/H5N1 avian influenza in domestic ferrets kept as pets. In addition, this outbreak suggests the possibility of asymptomatic A/H5N1 virus shedding by ferrets, highlighting their zoonotic potential and the advisability of excluding fresh or frozen poultry from their diet to reduce the A/H5N1 virus transmission risks.

## 1. Introduction

In June and July 2023, numerous deaths of cats with acute respiratory, neurological, and gastrointestinal symptoms were seen in various locations throughout Poland. These cases resulted from infection with the highly pathogenic avian influenza (HPAI) virus A/H5N1 [1,2,3,4]. Although A/H5N1 virus infections have been occasionally observed in cats in the past [5], the scale of these outbreaks in Poland turned out to be unprecedented, both in the European and the global context, which was reflected in the WHO announcement of 16 July 2023 (https://www.who.int/emergencies/disease-outbreak-news/item/2023-DON476 accessed on 5 June 2024). In total, the A/H5N1 virus infection was confirmed in at least 34 cats. The genomes of 19 virus isolates were sequenced and analysed, showing that all of them were closely related, belonging to the H5 clade 2.3.4.4b, and were similar to viruses which had been circulating in wild birds and poultry at that time in Poland [2,3]. An A/H5N1 virus infection was also confirmed during this time in a dog [6].

Both the American mink (*Neovison vison*) and the domestic ferret (*Mustela furo*) are highly susceptible to influenza A virus infections [7], making the latter a commonly used model for studying the pathogenicity, progression, and transmission of these viruses [8,9,10,11]. The first reported natural cross-species transmission of influenza from birds to minks, resulting in a clinical disease, occurred in 1984 across thirty-three mink farms along Sweden’s eastern coast. In October 2022, an outbreak of A/H5N1 clade 2.3.4.4b HPAI was reported on a Spanish mink farm, coinciding with an ongoing epidemic in birds. A subsequent HPAI outbreak in mid-July 2023 in Finland involved infections among farmed foxes, American minks, and raccoon dogs, with genetic analyses suggesting virus transmission from wild birds [12,13].

Though countless reports on experimental influenza infections in ferrets have been published [10,11,14,15], descriptions of natural A/H5N1 virus infections in this species are hardly to be found. In this report, we present such infections in clinically ill as well as asymptomatic domestic ferrets that occurred during concurrent outbreaks of HPAI in Polish cats in 2023.

## 2. Materials and Methods

### 2.1. Animals, Clinical Course and Treatment

A group of five co-housed ferrets, consisting of three juveniles (9 weeks of age), their mother (2 years and 7 months) and a second adult (3 years), were admitted on 26 June 2023 to a veterinary clinic in south Poland. All juveniles displayed significant lethargy and a pronounced unwillingness to move. All ferrets were previously fed commercially available poultry meat. According to the owner, the first symptoms of lethargy appeared in the juvenile ferrets approximately 12–14 h after the last meal. However, it was revealed that the juvenile ferrets were fed raw poultry meat from a different batch than both adults, that did not show any clinical abnormalities neither on presentation nor later. During clinical examination, exacerbated respiratory sounds were detected over both the upper and lower respiratory tracts in the three sick ferrets, with two of them presenting with tachypnoea. Upon palpation, their intestines appeared distended and filled with gas. The mucous membranes were uniformly pale pink across the group, with no indications of dehydration. A dental examination did not reveal any abnormalities, and there was no nasal discharge.

Due to signs of respiratory infection, the three affected ferrets received sulfamethoxazole with trimethoprim (Bactrim, Roche, Warsaw, Poland) at a dosage of 0.15 mL, administered twice daily. In addition, to reduce abdominal bloating, simethicone (Bobotic, Polpharma, Starogard Gdański, Poland) was given orally at a dosage of 0.3 mL three times a day. The owner presented the animals for the next time as late as after a week, with information that one ferret succumbed, presenting with dyspnea and neurological symptoms. The other two had no fever, but continued to show weakness, and pelvic limb ataxia was seen. Exacerbated respiratory sounds were less severe in them, but shortness of breath was still present. Therefore, 0.5 mL of gentamycin solution (40 mg/mL) (Gentamycin Krka, Krka-Polska, Warsaw, Poland) was mixed with 5 mL of physiological saline and delivered to the 2 ferrets via a nebulizer twice daily. In one of these ferrets, dyspnea and incoordination of the pelvic limbs persisted till the 11th day of treatment, but eventually it recovered completely. The second one also recovered fully within a few days after the last visit.

As at that time it was already known that a series of deaths among Polish cats with severe respiratory and neurological symptoms were caused by the A/H5N1 influenza virus, point-of-care tests for the type A influenza virus antigen were performed with throat swabs of all four alive ferrets. As the test results were positive, throat swabs were taken from all ferrets to confirm the antigen test result and to determine the subtype of the virus by RT-qPCR. Necropsy, histopathological and virological examinations were performed on the carcass of the succumbed ferret.

### 2.2. Anatomopathological Examination

Standard necropsy protocols were followed during the necropsy of the succumbed ferret. The carcass of the deceased ferret was maintained at −20 °C from the time of death until the necropsy and sample collection.

### 2.3. Histopathological Examination

For histological examination, samples from the brain, lungs, heart, liver, spleen, pancreas, kidneys, mesenteric lymph nodes, and intestines were preserved in 4% formaldehyde solution buffered with phosphate. Subsequently, the specimens were washed extensively under tap water, subjected to a graded dehydration process with ethanol and xylene, and finally embedded in paraffin wax. Sections of 4 µm thickness were obtained from these paraffin blocks, which were then stained using hematoxylin and eosin. The stained sections were evaluated under a light microscope (Olympus CX21, Olympus Corporation, Tokyo, Japan).

### 2.4. Virological Tests

Throat swabs for virological testing were collected during the clinical examination. Initial screening of the swabs was performed using the CorDx Influenza A/B + COVID-19/RSV Combo Ag (CorDx Inc., San Diego, CA, USA) test for the detection of viral antigens.

Next, molecular diagnostics for influenza virus type A (IAV) subtypes H1N1, H3N2, H5N1, influenza virus type B (IBV) and SARS-CoV-2 were performed. Using the Total RNA Mini Kit (A&A Biotechnology, Gdańsk, Poland), in accordance with the manufacturer’s instructions, RNA was extracted from intravitally collected throat swabs from four ferrets and from each 50 mg of fresh lung, trachea, brain, liver, intestine, heart, pancreas and kidney collected during the necropsy of the succumbed ferret.

One-step reverse transcription real-time PCR (RT-qPCR) was performed using the CFx96 system (BioRad, Hercules, CA, USA). Tests for SARS-CoV-2 were conducted with a commercially available Novel Coronavirus (COVID-19) Real-Time Multiplex RT-PCR Kit (LifeRiver, San Diego, CA USA), and tests for IAVs were performed with an in-house method described by Stefańska et al. [16].

Briefly, the specificity of qPCR reactions was checked using RNA isolated from various reference strains of influenza virus subtypes A(H5N1) as a template. RNA isolated from the uninfected MDCK cell line was used as a negative control (non-template control; NTC) for the reaction. To determine the limit of detection (LOD) of the method and the range of its linearity, we used pBluescript plasmids with a cloned cDNA fragment (385 nucleotides) covering the HA gene region (Epoch LifeScience^®,^ Missouri City, TX, USA), for which primers and a probe were previously designed. LOD was determined as the lowest cDNA concentration that would be detected with 95% probability when testing at least 24 samples constituting replicates of 5 dilutions of plasmid DNA on a decimal logarithmic scale [16]. RT-qPCR results with a quantification cycle (Cq) of ≤35.00 were considered positive.

## 3. Results

### 3.1. Anatomopathological and Histopathological Examination

During the cranial dissection, it was noted that the brain did not have its typical consistency. Upon opening the cranial structures, the cerebral material exhibited a consistency more akin to a semi-fluid than a solid. The remaining organs, namely, lungs, heart, liver, spleen, pancreas, kidneys, and intestines, did not show any macroscopic abnormalities, maintaining their original shape and texture.

The heart, brain and intestine wall had normal histological appearances with no pathological lesions. In contrast, the liver, lungs, and kidneys were congested. In addition, lung edema and parenchymatous degeneration of the renal tubules were found. However, it should be noted that histopathological changes within the lung tissue were the most expressed and are presented in Figure 1.

### 3.2. Virological Tests

The antigenic CorDx Influenza A/B + COVID-19/RSV Combo Ag test revealed orthomyxovirus type A antigens in all four throat swabs.

In the RT-qPCR, all throat swabs as well as samples from the lungs, trachea, heart, brain, pancreas, liver, and intestine wall of the succumbed ferret were found to be positive for A/H5N1 virus RNA, whereas the sample from the kidney was negative. Exact results of the RT-qPCR are shown in Table 1. All samples were RT-qPCR negative for all other tested viruses, namely, A/H1N1, A/H3N2, IBV, and SARS-CoV-2.

## 4. Discussion

Ferrets, due to their high susceptibility to influenza virus infections, serve worldwide as an established research model for exploring the pathogenesis of influenza [8,9,10,11]. Also, HPAI A/H5N1 virus infection has been extensively studied on experimental ferrets which have developed respiratory and neurological symptoms, severe lethargy, fever, weight loss, transient lymphopenia, and occasionally, digestive problems [8,17,18]. The clinical symptoms seen in the three clinically ill ferrets in our outbreak were concordant with the clinical picture after experimental infections, and in one of these ferrets, the disease was lethal. However, the natural outbreak of HPAI A/H5N1 virus infection in ferrets described in this paper brings to light several novel concerns.

First of all, the course of the infection differed significantly between adult and young animals. The two adults showed no clinical symptoms despite significant viral loads in the throat (Table 1). In contrast, nine-week-old ferrets showed primarily severe respiratory distress, accompanied by neurological symptoms. A similar situation occurred during the outbreak caused by the HPAI A/H5N1 virus on fur farms in Finland in July 2023, when mainly young blue and silver foxes, raccoon dogs and American minks were affected [13]. These differences could probably be partially explained by different immunological experiences and the degree of development of the immune system in juveniles and adults. However, in the outbreak described in the present paper, young ferrets were fed with poultry meat from a different batch than the adults. Hunting infected birds and eating raw contaminated poultry products are typical reasons for HPAI cases in cats and other carnivores [19]. Infection through contaminated poultry meat was also suspected in feline cases in Poland 2023 [3], and the ferret outbreak occurred at the same time. Though other routs cannot be excluded, for example, the owners’ shoes contaminated with faeces of infected birds, the alimental infection is much more probable, as the affected ferrets were kept strictly indoors, and fed raw poultry meat. Therefore, the questions arise whether (1) young ferrets were infected after consuming poultry meat and then infected the adults, or (2) all ferrets were infected consuming poultry meat? Although the mammal-to-mammal transmission of A/H5N1 virus has not yet been strictly proven, several instances suggest the possibility of such an event. Recently, in experimental conditions, co-housing of infected and non-infected ferrets promoted the transmission of the virus between these two groups [18]. Moreover, transmission between mammals was suggested during an A/H5N1 influenza outbreak on a mink farm in Spain and on multiple fur farms in Finland [12,13]. Similarly, during an outbreak of A/H5N1 HPAI in Thailand, horizontal transmission among tigers was suspected in a zoo [20]. Also, the scale of the mass mortality in South American sea lions raised speculations about the transmission of the A/H5N1 virus infection between the sea lions themselves [21,22]. Therefore, in the described five co-housed ferrets, the first proposed transmission scenario, though unverified, warrants consideration, especially since it is also known that the course of the A/H5N1 virus infection in ferrets depends not only on the immune status of the animals, but also on the route of infection [17].

Another concerning aspect is the high susceptibility of ferrets to various influenza A virus subtypes, potentially allowing for co-infection with multiple subtypes. Such scenarios could lead to the emergence of new influenza virus reassortants with enhanced human cell affinity and maintained high pathogenicity. Given the growing popularity of ferrets as pets globally, and their close contact with humans, this could contribute to an adaptation of this virus to humans.

Cases like these highlight the imperative need for vigilant HPAI surveillance also in ferrets and a deeper understanding of the pathogenesis, transmission dynamics, and cross-species infectivity potential of the A/H5N1 virus. Two of the five infected ferrets remained clinically healthy while RT-qPCR confirmed A/H5N1 RNA in their throat swabs (Table 1). Subclinical HPAI A/H5N1 virus infections are possible also in birds, especially wild ones, but were documented also in cats [23]. Asymptomatic infections documented in two adult ferrets mean that clinically healthy animals, maintaining close contact with their owners without raising any suspect, could be shedders of the virus. This point is especially significant given the zoonotic potential of the A/H5N1 virus. Human infections are extremely rare, but can be very severe, and the case fatality rate is over 50% [24].

## 5. Conclusions

This outbreak showed that ferrets naturally infected by the HPAI A/H5N1 virus can either die, or remain asymptomatic, while having a high viral load in the respiratory tract. To our best knowledge, this is the first documented natural infection with HPAI A/H5N1 virus in ferrets kept as pets.

## Figures and Tables

**Figure 1 viruses-16-00931-f001:**
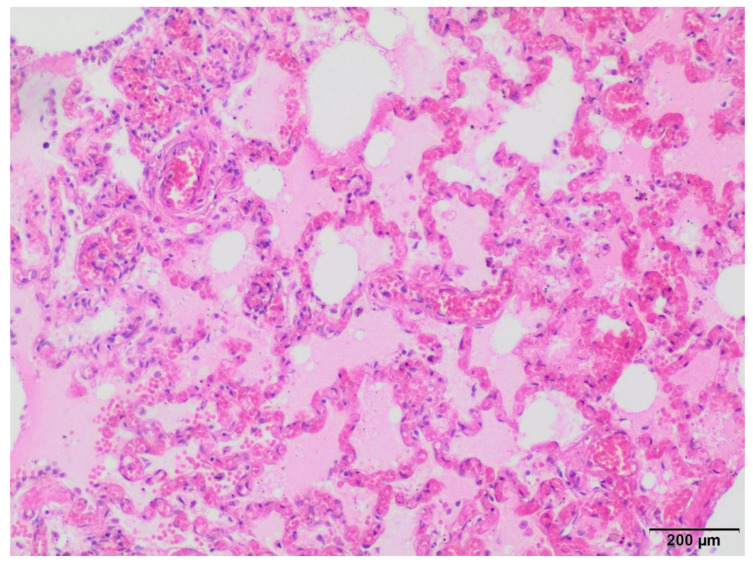
Microscopic appearance of lung sample from ferret lung congestion and edema are present; hematoxylin–eosin stain, 100× magnification.

**Table 1 viruses-16-00931-t001:** Results of RT-qPCR for influenza A/H5N1 virus gene. Ct value > 35.00 was considered negative.

Number	Age	Laboratory Number	Type of Sample	Result	Ct for H5 Gene	Ct for N1 Gene
1	Adult (mother)	F5	pharyngeal swab	(+)	30.21	31.02
2	Adult	F4	pharyngeal swab	(+)	27.14	27.32
3	Young	F3	pharyngeal swab	(+)	34.78	34.90
4	Young	F2	pharyngeal swab	(+)	28.54	28.72
5	Young	F1	lung	(+)	25.62	26.03
6	Young	F1	liver	(+)	31.54	32.00
7	Young	F1	spleen	(+)	32.04	32.31
8	Young	F1	intestine	(+)	34.23	34.71
9	Young	F1	brain	(+)	34.55	34.83
10	Young	F1	heart	(+)	34.12	34.62
11	Young	F1	trachea	(+)	34.17	34.88
12	Young	F1	pancreas	(+)	30.74	31.10
13	Young	F1	kidney	(−)	-	-

## Data Availability

No new data were created or analysed in this study. Data sharing is not applicable to this article.

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
