# Peer review of "Natural Infection with Highly Pathogenic Avian Influenza A/H5N1 Virus in Pet Ferrets"

_viruses, 2024, doi:10.3390/v16060931_

Round 1

Reviewer 1 Report

Comments and Suggestions for Authors

In this brief article by Golke et al, authors describe infection of 5 pet ferrets with H5N1 infection in Poland in 2023. Authors report clinical illness, histopathological findings, and PCR-positive detection in swabs and postmortem tissues from animals. While the source of infection and potential transmission of virus between animals is unconfirmed, the data are nonetheless important to report. However, while authors report PCR data, no histopathological images are shown to support what is stated in the text.

 Major comments:

 -Sections 3.1 and 3.2 of results – these sections could be combined as both are very brief and underdeveloped. For section 3.1, “remaining organs” is not specific; please specify which organs did/did not appear normal. For section 3.2, please show representative histopathological results in a figure to support text describing the findings.

 -Table 1. How was the F1 ferret that succumbed to infection stored after death (as the text implies the animal died outside of a veterinary clinic) and how much time passed between animal death and collection of specimens for RT-PCR assessment?

 Minor comments:

 -throughout the manuscript, please change ‘patients’ to a different descriptor (e.g. ‘animals’).

 -please provide source of raw poultry meat that juvenile ferrets were fed if known, and provide text to state how long between post-feeding of raw poultry meat and first symptom onset of the juvenile ferrets (in section 2.1).

 -please state the approximate age of ferrets F4 and F5 (text just states ‘adult’ but it is unclear what potential age range this descriptor encompasses).

 -please specify limit of detection for calling a negative result in a footnote in Table 1.

Author Response

Reviewer 1: In this brief article by Golke et al, authors describe infection of 5 pet ferrets with H5N1 infection in Poland in 2023. Authors report clinical illness, histopathological findings, and PCR-positive detection in swabs and postmortem tissues from animals. While the source of infection and potential transmission of virus between animals is unconfirmed, the data are nonetheless important to report. However, while authors report PCR data, no histopathological images are shown to support what is stated in the text.

 Major comments:

 -Sections 3.1 and 3.2 of results – these sections could be combined as both are very brief and underdeveloped. For section 3.1, “remaining organs” is not specific; please specify which organs did/did not appear normal. For section 3.2, please show representative histopathological results in a figure to support text describing the findings.

Our response: As suggested, sections 3.1 and 3.2 have been combined and expanded, with more detailed information included. Histopathological changes observed in the lungs of a deceased ferret are shown in Figure 1.

Reviewer 1: Table 1. How was the F1 ferret that succumbed to infection stored after death (as the text implies the animal died outside of a veterinary clinic) and how much time passed between animal death and collection of specimens for RT-PCR assessment?

Our response: Information regarding the storage of dead ferret carcasses has also been added in section 2.2. Due to the numerous H5N1 infections in cats in Poland at that time, clinical symptoms, and a positive rapid test for influenza A, the owner was informed about the risk of H5N1 infection in ferrets and agreed to submit the carcass for further examination in the event of the animal's death. The owner delivered the carcass of the deceased ferret in a frozen form, and the time between death and sample collection was approximately 1 week. At that time, the carcass was stored at a temperature of -20°C.

Reviewer 1:  Minor comments:

 -throughout the manuscript, please change ‘patients’ to a different descriptor (e.g. ‘animals’).

 -please provide source of raw poultry meat that juvenile ferrets were fed if known, and provide text to state how long between post-feeding of raw poultry meat and first symptom onset of the juvenile ferrets (in section 2.1).

 -please state the approximate age of ferrets F4 and F5 (text just states ‘adult’ but it is unclear what potential age range this descriptor encompasses).

 -please specify limit of detection for calling a negative result in a footnote in Table 1.

Our response: Done. Information on the age of the ferrets, the origin of the poultry meat, and the time between eating a meal and the onset of clinical symptoms has been added to the text. All changes have been highlighted.

Reviewer 2 Report

Comments and Suggestions for Authors

This paper presents a case report on H5N1 infection in ferrets. The available data consist solely of CT values for the H5 HA and NA genes. However, the qRT-PCR lacks both positive and negative controls. The methods section states, "RT-qPCR results with a quantification cycle (Cq) of ≤ 35.00 were considered positive." It is unclear whether this threshold is supported by pilot studies, previous research, or other publications. Additionally, no histopathological examination figures are provided. Overall, the data and case description are insufficient and do not convincingly demonstrate that this is an H5N1 infection case.

Author Response

Reviewer 2: This paper presents a case report on H5N1 infection in ferrets. The available data consist solely of CT values for the H5 HA and NA genes. However, the qRT-PCR lacks both positive and negative controls. The methods section states, "RT-qPCR results with a quantification cycle (Cq) of ≤ 35.00 were considered positive." It is unclear whether this threshold is supported by pilot studies, previous research, or other publications. Additionally, no histopathological examination figures are provided. Overall, the data and case description are insufficient and do not convincingly demonstrate that this is an H5N1 infection case.

Our response: We would like to thank the reviewer for drawing our attention to a possible lack of clarity in the description of the methodology used. All investigations were conducted precisely according to the procedure outlined by Stefańska et al., as cited below, and included both non-template controls (NTC) and positive controls containing genetic material isolated from influenza virus laboratory strains. The cycle threshold (Ct) set at ≤ 35.00 cycles is a typical quantification limit, adopted in many commercial qPCR tests and "in-house" methods developed in laboratories. This value approximately corresponds to several dozen viral copies per milliliter, which is the detection limit for most qPCR assays.

We have added information about the qPCR method we use to the manuscript.

A figure showing histopathological changes in the lung tissue of a dead ferret was also added to the manuscript.

Reviewer 3 Report

Comments and Suggestions for Authors

This case report describes the natural infection of ferrets with avian influenza H5N1 after exposure to raw poultry meat. The paper is succinct and well-written, while providing an appropriate description of the clinical case.

It is interesting that the one ferret that passed away presented with no significant gross lesions. The deceased ferret presented with the highest viral load of all specimens tested, which is noteworthy and consistent with the clinical observations.

I wonder if the owners could have been infected and asymptomatic in this case too, since these ferrets were shedding the virus for such a long period of time. Like the authors mentioned, it would be an opportunity for recombination events between HPAI H5N1 and human influenza A viruses.

I would add in the discussion a paragraph about the relevance of raw poultry meat in the transmission of avian influenza to mammals. Are there any reports of human infection with avian flu transmitted from the handling of raw meat? Cross-contamination in the kitchen is common and a major concern for food-borne diseases, I wonder if avian flu should also be considered a risk.

Author Response

Reviewer 3: This case report describes the natural infection of ferrets with avian influenza H5N1 after exposure to raw poultry meat. The paper is succinct and well-written, while providing an appropriate description of the clinical case.

It is interesting that the one ferret that passed away presented with no significant gross lesions. The deceased ferret presented with the highest viral load of all specimens tested, which is noteworthy and consistent with the clinical observations.

I wonder if the owners could have been infected and asymptomatic in this case too, since these ferrets were shedding the virus for such a long period of time. Like the authors mentioned, it would be an opportunity for recombination events between HPAI H5N1 and human influenza A viruses.

I would add in the discussion a paragraph about the relevance of raw poultry meat in the transmission of avian influenza to mammals. Are there any reports of human infection with avian flu transmitted from the handling of raw meat? Cross-contamination in the kitchen is common and a major concern for food-borne diseases, I wonder if avian flu should also be considered a risk.

Our response: According to the current state of knowledge, consumption of meat from infected birds is the primary route of transmission of A/H5N1 viruses to carnivorous animals. As we noted in the discussion section: Hunting infected birds and eating of raw contaminated poultry products is a typical reason of HPAI cases in cats and other carnivores [19]. Infection through contaminat-ed poultry meat was also suspected in the feline cases in Poland 2023 [3].

However, the route of infection can only be suggested and in practice it is unfortunately impossible to determine. For this reason, the authors prefer to refrain from speculation.

Similarly, it is extremely difficult to predict the extent to which potentially A/H5N1-contaminated poultry meat may pose a risk to humans. The key issues here will likely be the infectious dose and the individual's immune status. To the authors' knowledge, no direct link between participation in the processing of raw poultry meat and human infections has been demonstrated so far. Such a source of infection remains hypothetical and has not been confirmed in publications or other reports. Nevertheless, the latest human fatality with this virus in Vietnam may indicate a link between hunting of wild birds and infection, as described in the WHO announcement.

“On 25 March 2024, Viet Nam National Focal Point (NFP) for International Health Regulations (IHR) notified the World Health Organization (WHO) of one case of human infection with an influenza A(H5N1) virus in a 21-year-old male with no underlying conditions from Khanh Hoa Province, Viet Nam. The case developed a fever and cough on 11 March 2024 and was admitted on 15 March to a local hospital due to persistent symptoms, including abdominal pain and diarrhoea. The case died on 23 March. Initial results from the case investigation revealed that during the second and third weeks of February 2024, the case went bird hunting. Between that time and the onset of illness, no contact with dead or sick poultry nor contact with anyone exhibiting similar symptoms was reported.”

https://www.who.int/emergencies/disease-outbreak-news/item/2024-DON511 

In conclusion, we sincerely hope that our explanations and the revisions made will meet the reviewers' expectations. We trust that the changes enhance the clarity and quality of our manuscript. We appreciate your consideration and look forward to your positive decision regarding the publication of our article.

Round 2

Reviewer 2 Report

Comments and Suggestions for Authors

The authors updated some information according to what I mentioned in the last review. I still have concerns about whether these data can confirm it is an H5N1 infection case. Here, I have three more comments:

1. Can you also list the CT values of the positive and negative controls in one table along with the specimens' data?

2. Did you test the antibody titers in the serum of the recovered ferrets, especially against H5 viruses?

3. Did you amplify whole or partial viral genes and obtain some sequence information? It would be best if you successfully isolated the H5 virus.

Comments on the Quality of English Language

Overall, I still have concern if the data can convincingly demonstrate that this is an H5N1 infection case.

Author Response

Reviewer: The authors updated some information according to what I mentioned in the last review. I still have concerns about whether these data can confirm it is an H5N1 infection case. Here, I have three more comments:

  1. Can you also list the CT values of the positive and negative controls in one table along with the specimens' data?
  2. Did you test the antibody titers in the serum of the recovered ferrets, especially against H5 viruses?
  3. Did you amplify whole or partial viral genes and obtain some sequence information? It would be best if you successfully isolated the H5 virus.

Overall, I still have concern if the data can convincingly demonstrate that this is an H5N1 infection case.

Dear Reviewer,

Thank you for your comments and for the opportunity to address your concerns regarding our confirmation of H5N1 infection in ferrets using the RT-qPCR method according to Stefanska et al.

We would like to assure you that the RT-qPCR protocol we employed is rigorously validated and widely recognized for its specificity and sensitivity in detecting H5N1. Our laboratory follows stringent quality control measures to ensure the accuracy and reliability of our results. The primers and probes used in our RT-qPCR assays were designed based on sequences highly conserved among H5N1 strains, as detailed in Stafanska et al.'s publication.

In our study, multiple replicates of each sample were tested to confirm the presence of H5N1 RNA. Positive and negative controls were included in each run to monitor for potential contamination and to validate the amplification process. Furthermore, the Ct values obtained for our positive samples were well within the expected range, providing additional confidence in the specificity of our assay.

Additionally, our authors have extensive experience in influenza diagnosis using the RT-qPCR method and the development of qPCR tests used in virological diagnostics, supported by numerous publications, and RT-qPCR is a well-established and recognized method for influenza diagnosis.

Finally, please consider that influenza has been recognized not only basing solely on the RT-qPCR result. According to the medical principles, also this diagnosis has been made considering in addition all relevant clinical, epidemiological and laboratory findings:

  • Presence of orthomyxovirus type A antigen in the throat swabs.
  • Absence of RNA of some other viruses (A/H1N1, A/H3N2, IBV, and SARS-CoV-2) that could induce a similar respiratory disease.
  • Typical risk factor of A/H5N1 virus infection in cats and other carnivores (eating of uncooked poultry products).
  • Epidemiological situation (at that time in our country 3 independent, scientific groups found fatal A/H5N1 virus infections in cats, and one of these groups found A/H5N1 virus in poultry meat available for sale).
  • Very short incubation period and very acute clinical course.
  • Typical clinical signs (respiratory distress and neurological signs).

-      Pathological findings in the brain and lungs consistent with those typical for     A/H5N1 virus infection in cats and other carnivores.

Ad. 1) We can provide Ct values to both controls of the reaction, although this is not used in scientific papers. The negative control was/is assumed to be negative, so in this case the value is 0.00.

Ad. 2) As the tested ferrets, including the recovered one, came from a private owner, we were not able to collect additional serum samples. Commercial ELISA tests known to us allow the detection of antibodies against only of type A and/or B influenza viruses, but not the presence of antibodies anti-H5. The seroneutralization test or hemagglutination inhibition assay used for this purpose were not the aim of this study.

Ad.3) We did not have the opportunity to sequence the collected influenza isolates. The primary reason for this was needed to work in the BSL-3 laboratory, related to the isolation of the H5N1 influenza virus. The second one was focus of our study on RT-qPCR methods, as an element of routine diagnostics. Products amplified by RT-qPCR using TaqMan probes typically amplify gene fragments of lenght 130-150 bp. Therefore, it is not possible to amplify the whole gene using this method.  At the same time, sequencing is not an element of routine diagnostics of influenza viruses, both in medical and veterinary virology.
